# Response to Static Magnetic Field-Induced Stress in *Scenedesmus obliquus* and *Nannochloropsis gaditana*

**DOI:** 10.3390/md19090527

**Published:** 2021-09-21

**Authors:** Génesis Serrano, Carol Miranda-Ostojic, Pablo Ferrada, Cristian Wulff-Zotelle, Alejandro Maureira, Edward Fuentealba, Karem Gallardo, Manuel Zapata, Mariella Rivas

**Affiliations:** 1Laboratorio de Biotecnología Ambiental Aplicada, Departamento de Biotecnología, Universidad de Antofagasta, Av. Angamos 601, Antofagasta 1270300, Chile; genesis.lsd1@gmail.com (G.S.); carol.miranda@ua.cl (C.M.-O.); alejandro.maureira.calderon@ua.cl (A.M.); manuel.zapata@uantof.cl (M.Z.); 2Centro de Desarrollo Energético Antofagasta, Universidad de Antofagasta, Av. Angamos 601, Antofagasta 1270300, Chile; edward.fuentealba@uantof.cl; 3Laboratorio de Biología Celular, Molecular y Genética, Departamento Biomédico, Universidad de Antofagasta, Av. Angamos 601, Antofagasta 1270300, Chile; cristian.wulff@uantof.cl; 4Centro de Investigación Tecnológica de Agua en el Desierto (CEITSAZA), Universidad Católica del Norte, Av. Angamos 0610, Antofagasta 1270709, Chile; kgallardo@ucn.cl

**Keywords:** enzymatic activity, fluid dynamics, microalgae, oxidative stress, static magnetic fields, violaxanthin

## Abstract

Magnetic fields in biological systems is a promising research field; however, their application for microalgae has not been fully exploited. This work aims to measure the enzymatic activity and non-enzymatic activity of two microalgae species in terms of superoxide dismutase (SOD), catalase (CAT), and carotenoids, respectively, in response to static magnetic fields-induced stress. Two magnet configurations (north and south) and two exposure modes (continuous and pulse) were applied. Two microalgae species were considered, the *Scenedesmus obliquus* and *Nannochloropsis gaditana*. The SOD activity increased by up to 60% in *S. obliquus* under continuous exposure. This trend was also found for CAT in the continuous mode. Conversely, under the pulse mode, its response was hampered as the SOD and CAT were reduced. For *N. gaditana*, SOD increased by up to 62% with the south configuration under continuous exposure. In terms of CAT, there was a higher activity of up to 19%. Under the pulsed exposure, SOD activity was up to 115%. The CAT in this microalga was increased by up to 29%. For *N. gaditana*, a significant increase of over 40% in violaxanthin production was obtained compared to the control, when the microalgae were exposed to SMF as a pulse. Depending on the exposure mode and species, this methodology can be used to produce oxidative stress and obtain an inhibitory or enhanced response in addition to the significant increase in the production of antioxidant pigments.

## 1. Introduction

There are several cases where magnetism is present in living matter. For instance, the magnetotactic bacterium can align its body according to the Earth’s magnetic field. This feature is possible due to chains of single-domain, biogenic magnetite (Fe_3_O_4_) able to sense geomagnetic fields [1]. Macroscopic organisms, such as birds, are known to orient themselves to magnetic field lines and to travel long distances [2]. In fact, all matter falls into one of these categories: diamagnetism, paramagnetism, ferromagnetism, ferrimagnetism or antiferrimagnetism [3]. Firstly, magnetic fields (MF) conceived as a perturbation, which can be used by organisms in a natural way, or MF applied to biological systems to promote changes at several levels, can be classified according to the broader and general perspective of electromagnetic (EM) fields. The detection/sensing or the response to MF can be associated with different mechanisms occurring at the tissue, cell, membrane, and molecular levels, including the medium where the organisms live [4]. To understand interaction mechanisms and effects of static magnetic fields (SMF) on cells, a categorization of EM fields is needed and provided in the following.

Electromagnetic fields (EM) can be classified according to four main parameters: time-dependence, distribution or homogeneousness, direction, and exposure time. The first parameter can be split into two branches, namely, time-constant (static EM fields) or time-varying (dynamic EM fields). The static EM fields are grouped as stationary fields not exhibiting mutual coupling between the electric and magnetic component.

Regarding the second parameter for classification, the distribution of the fields can be homogeneous when same magnitude occurs in all space or inhomogeneous when the magnitude varies with one or more than a spatial coordinate. The third parameter refers to the direction of the field. When implementing an array of electro or permanent magnets, depending on location and orientation, many patterns and intensities are possible [5]. The fourth parameter refers to how much time a specimen or biological system is exposed to the MF. Different processes within a biological system can occur at times, which can differ in orders of magnitude [6].

Parameters such as the magnet materials used, magnet dimensions, pole configuration, measurement of the field strength, frequency of application, duration and site of application, magnet support device, target tissue, and distance from magnet surface have been discussed by Colbert et al. [7]. These authors performed an analysis based on 56 articles focusing on the quality of reporting SMF and those parameters concluding that 61% of the studies failed to provide enough experimental details regarding SMF. Consequently, the possibility for other researchers to replicate protocols and give satisfactory explanations might be reduced. These findings highlight the importance of providing proper conditions for SMF exposure and characterization.

Regarding effects of SMF on different cell parameters, authors found that cell size, shape, orientation, and membrane roughness changed due to the exposure [6]. Researchers showed an increased number and size of holes in cell membrane [8]. Thus, the volume force due to magnetic gradient led to higher membrane permeability. The magnetic gradient just mentioned was examined by Zablotskii et al. [9] where the discussion was centered on the mechanism by which MFs act on tissues. It was argued that it is the high magnetic gradient and not the field itself which can cause several effects in cells and their functions. They showed that high MF gradient can be involved in the change of the ion-channel on/off probability, suppression of cell growth by magnetic pressure, magnetically induced cell division and cell reprograming, and forced migration of membrane receptor proteins. Fundamentals regarding how MF can alter the rate, yield, and product distribution of chemical reactions can be seen in terms of magnetokinetics [10]. A review on the effects of SMF on cellular systems provides a wide insight regarding the most influential factors and mechanisms [11]. Authors point out that the radical pair recombination and the diamagnetic anisotropy can cause changes and subsequent effects on the susceptibility of biomolecules, intracellular structural modifications, and changes in the enzymatic reactions. Some studies have used SMF to control algal growth [12], and changes in biochemical composition and photosynthesis have also been described [12]. In contrast, other studies propose their use as biostimulants for growth [13]. It has been shown that SMF promotes growth and oxygen production in *Scenedesmus obliquus* [14]. Moreover, an increase in growth (100%), stimulation of antioxidant defense in *Chlorella vulgaris* has been demonstrated [15], and production of exopolysaccharides [16].

The main cellular physiological processes generate reactive oxygen species (ROS). To classify ROS into categories, there are free radicals and non-radical species [17]. The radicals are understood as molecular species, which contain an unpaired electron in an atomic orbital [18]. Consequently, a free radical tends to remove an electron from a stable molecule to reach electrochemical stability. Accordingly, oxygen, hydrogen atom, and transition metals constitute free radicals [17]. Particularly, this unpaired electron makes these species paramagnetic and susceptible to magnetic fields. In the case where the field is present in biological systems, it may affect ROS dynamics. Particularly, in the photosynthetic cells as the case of microalgae, the main organelles generating ROS are mitochondria, chloroplasts, and to a lesser extent, peroxisomes, where they are formed as a consequence of oxidative energy metabolism and the intense flow of electrons that exists in the mitochondrial and thylakoidal membranes [19]. The ROS are highly reactive and can oxidize essential macromolecules. All cells possess an antioxidant defense system that neutralizes or metabolizes ROS. This system includes antioxidant enzymes (catalase, superoxide dismutase, ascorbate peroxidase, glutathione peroxidase, among others) and low molecular weight metabolites (glutathione, ascorbic acid, carotenes, phenols, flavonoids) [20]. If this system of defenses is overcome, the cellular oxide-reduction equilibrium shifts to a pro-oxidant situation and a state of oxidative stress is reached in which oxidative damage occurs to proteins, nucleic acids, and lipids [21]. In addition, various environmental factors and the entry of contaminants into the cells can intensify the production of these molecules; the most harmful being the superoxide anion (O_2_^●−^), hydrogen peroxide (H_2_O_2_), and hydroxyl radical (OH^−^) [22,23].

The non-enzymatic antioxidant system includes compounds having a low molecular weight, where the most important are the reduced glutathione, vitamin E, vitamin C, such as ascorbic acid, and vitamin A. In addition, there are flavonoids, phenolic acids, α-lipoic acid, uric acid, bilirubin, some sugars, and amino acids. In the defense mechanism, these compounds capture free radicals avoiding chain reactions. They can be associated with the membrane (vitamin E, carotenoids, phycobiliproteins), dispersed in the cytoplasm (mycosporine-like amino acids), or linked to water-soluble reducing compounds, such as GHS and vitamin C [24,25].

In the present work, we investigated the influence of SMF on the response of two microalgae, growing in different culture media. The model species used were the *S. obliquus* and the *Nannochloropsis gaditana*, as they have known structure, functionality, and rapid growth. Furthermore, these species exhibit biotechnological potential in the generation of antioxidant biomolecules. *S. obliquus* is dominant in freshwater bodies, lakes, and rivers [26], it is cultivated at industrial scale to produce bioactive molecules, and under stress conditions it has been shown to accumulate β-carotene, astaxanthin isomers, lutein, and canthaxanthin [27,28]. Moreover, *N. gaditana* is known to produce several compounds such as fatty acids, lipids, including pigments such as zeaxanthin, astaxanthin, and canthaxanthin [28,29]. The focus of this work is the analysis of the oxidative stress of the selected species to deepen the understanding of the effects of SMF and consequently the potential production of antioxidant pigments. Nevertheless, the pigment composition of algae varies from species to species and according to environmental conditions including different stress conditions such as temperature, salinity, and irradiance, which may induce carotenoid production in algal species. Therefore, analysis of a complete pigment spectrum was intended for both microalgae in the control condition and treatments. For this purpose, we used two magnet configurations and two exposure modes and characterized the changes in terms of enzymatic activity and non-enzymatic production. We then discussed the results taking into consideration the characterization related to a single exposure mode previously reported in Ferrada et al. [30], and thus providing a broader analysis.

## 2. Results

The physicochemical medium parameters such as temperature (T), pH, electrical conductivity (*σ*), salinity (*S*), and dissolved oxygen (DO) were characterized for *S. obliquus* and *N. gaditana* for the control condition and under the influence of SMF (see Table 1 for a summary). Regardless of the species and treatment used, i.e., control or using a magnet configuration and exposure mode, T and pH remained at similar values in the range of 21–22 °C and 8.6–8.7, respectively. However, the electrical conductivity, salinity, and dissolved oxygen concentration were different depending on the medium. *N. gaditana* grows in a medium, in which *σ* is ~46 times higher and *S* is ~60 times higher than that of *S. obliquus.* Conversely, DO for *S. obliquus* is 20% higher than the medium for *N. gaditana.* Qualitatively, these results mean that the changes induced by SMF were not observed at the macroscopic level, that is, in terms of physicochemical parameters. This highlights the need to look inside the cell and quantify the oxidative stress, which is exposed in Section 2.1. A visualization of the magnetic flux density B is shown below.

The magnetic flux density norm (|B|) is visualized in Figure 1 corresponding to cases when all north or south poles are pointing towards the center of the flask. Figure 1A depicts circles to indicate where the field is specified (Figure 1B). Similarly, Figure 1C indicates where the field is computed (Figure 1D). The case of all north poles oriented to the center results in field lines which oppose each other and leads zero magnitude of B at the center of the flask at z = 0.

### 2.1. Response of Microalgae to SMF

*S. obliquus* and *N. gaditana* microalgae were exposed to SMF for 96 h and their growth was determined by optical density at 680 nm. The control condition corresponds to microalgae without exposure to SMF and in the presence of a SMF equal to 4500 G (referred as intensity at the side of the magnet, see Figure 1) with north and south arrangement in pulse or continuous form during 96 h. For *S. obliquus* after 48 h of growth, significant differences between the control and the exposed microalgae are evidenced, in the presence of SMF North-C and North-P a decrease was observed from 48 to 96 h (OD680 nm equal to 0.46 and 0.53, respectively) compared to the superior growth of the control condition with an OD680 nm equal to 0.95 (Figure 2A). On the contrary, in the presence of SMF South-C and P, less growth was observed in the microalgae exposed continuously at 48 h and 72 h (equal to OD680 nm of 0.52 and 0.71) (Figure 2C). For *N. gaditana*, clear effects of SMF on growth were observed from 48 h. Lower growth was observed in both treatments with north continuous and pulse exposure, which was more evident at 72 h or 96 h (Figure 2B). In response to exposure to south configuration, reduced growth is observed from 24 h of culture (Figure 2D).

Enzymatic activity was determined after 48 h of exposure to SMF. Figure 3 and Figure 4 show the measured activity of two antioxidant enzymes, the superoxide dismutase (SOD) and catalase (CAT), respectively, whereas Figure 5 depicts metabolites of interest. It is observed that the SMF influences the enzymatic activity in the *S. obliquus* since the SOD levels under continuous exposure increased up to 1.27 mUmg^−1^ for north and 1.41 mUmg^−1^ for the south configuration, representing a 65% and 42% relative difference with respect to the control condition. Nevertheless, the behavior is apparently reverted to an inhibitory effect under the pulse exposure mode since SOD activity with respect to the control values reduced by 26.3% for the north and 50.5% for the south magnet configurations, corresponding to 0.55 and 0.44 mUmg^−1^ of enzymatic activity (Figure 3A,B). In the case of *N. gaditana* SOD increased up to 0.94 mUmg^−1^ with the north and 1.20 mUmg^−1^ with the south magnet configuration under the continuous exposure mode, representing 38% and 62% higher production compared to the control values. Under the pulse exposure mode, SOD activity was 78% and 116% above that of the control for the north and south magnet configuration, respectively (equivalent to 1.2 mUmg^−1^ and 1.6 mUmg^−1^, respectively) (Figure 3C,D).

For *S. obliquus* the upward trend is also found for the CAT activity in the continuous mode combined with the south configuration, since there was a 37% increase in production compared to the control case (Figure 4B). The inhibition is also observed for the CAT activity under both exposure modes, since the reduction in CAT activity reached 14% for the north and 7.6% for the south magnet configuration (Figure 4A,B). On the other hand, for *N. gaditana* in terms of CAT, there was a higher activity expressed with a 19% and 9% higher production with respect to control values for the north south magnet configuration under continuous exposure mode, respectively (Figure 4C,D). For pulse exposure mode, CAT activity in this microalga reached 0.13 mU mg^−1^ for north and south configuration, meaning 29% and 23% with respect to their control conditions, respectively (Figure 4C,D).

We quantified metabolites of interest after 96 h of SMF exposure. Figure 5A,B indicates respectively that for *S. obliquus* there is a slight decrease in the production of violaxanthin in both exposure modes, a higher production of lutein under continuous exposure, but the same behavior as the control under pulsed exposure. In both cases the variation in the production of violaxanthin and lutein for the *S. obliquus* represents less than 1% with respect to the control. Conversely, Figure 5C reveals that for *N. gaditana*, violaxanthin increased by 10% under continuous SMF and 40% under pulsed exposure compared to the control. Figure 5D shows that the changes in zeaxanthin for *N. gaditana* with respect to the control are below 1%. These results highlight the effectiveness of the SMF on *N. gaditana*, especially for the pulse exposure mode.

## 3. Discussion

Magnetic fields are capable of inducing effects in many biological systems both in vivo and in vitro [12,13,14,15,16,17]. Some studies have been successful in stimulating the production of an antioxidant response to physiological stress induced by SMF or MF, as well as stimulating growth in microalgae such as *Chlorella vulgaris* [15,16] and *Dunaliella salina* [31]. However, the strength of the SMF may influence the effect on microalgae as described by Hunt et al. [32]. In contrast, in our study, only a significant increase in growth was observed for *S. obliquus* exposed to a north SMF in pulse form at 72 and 96 h. In all the other conditions analyzed, a decrease in growth or no clear effect on growth was determined, as was the case for *N. gaditana*. However, we note that in other studies the strength of SMF used is lower, for example, for *C. vulgaris* an increase in growth is described with exposure to a 1500 G SMF for 384 h, while high-intensity magnetic fields inhibit the growth of algal biomass [16].

The antioxidant enzymes SOD and CAT are characterized to present paramagnetic elements (Fe, Mn, and Cu) in their catalytic domain, and these could be strongly affected as consequence of the magnetic fields to which the algae were exposed during the experiments. From our study, it is possible to generalize that for *S. obliquus*, under the continuous exposure mode, there is a tendency for a higher production of SOD and CAT. Contrariwise, the pulse exposure mode led to an inhibitory effect on this microalga supported by a decrease in the SOD activity (20–50%) and CAT activity (10–20%). A possible explanation is that due to a saturation of ROS production inside the cell, it affects the metal ions present in the active site of the enzymes, preventing the substrate-enzyme assembly [33]. These results agree with those obtained by Corpas et al. [34], where the ROS production by lead in the *Arabidopsis thaliana* plant inhibits the catalase activity with respect to the control. A similar effect was observed by Chokshi et al. [35], where the salinity-induced oxidative stress in the *Acutodesmus dimorphus* microalgae alters the enzymatic activity in terms of SOD and CAT, as it was below the control values over time. Another point is that despite having a medium with a low electrical conductivity, interacting electrons in molecules are affected by magnetic fields, adding that the cell structure has a slightly rigid wall, allowing greater permeability to internal molecular changes [36]. This peculiarity in the functioning of enzymes was observed by Ferrada et al. [30], where an inhibition of enzymatic activity with respect to the control *in S. obliquus* exposed to SMF occurs, leading to state that the amount of ROS produced exceeds the threshold for the proper cells’ antioxidant capacity.

In the same way, the results for the *N. gaditana* allows for stating that, regardless of the exposure mode and magnet configuration, there is always a significant increase in the SOD and CAT production with respect to the control condition. These results also agree with Ferrada et al. [30], where an increase in the enzymatic activity in terms of SOD is observed in *N. gaditana* when exposed to SMF. Furthermore, in *C. vulgaris*, [16] demonstrated a higher induction of CAT activity at 101.37 Umg^−1^ using a 400 G SMF. However, as they increase the SMF intensity up to 1500 G, CAT activity decreases in agreement with the results observed in *S. obliquus* (Figure 4) exposed to approximately 4500 G in our study. Considering the SOD and CAT results, it is inferred that SMF impacts the antioxidant response of algae, although the effects on different enzymes and algae are inconsistent. It has been described that high magnetic field strengths produce changes in enzyme conformation, which in turn affects cellular biochemical reactions [37]. Furthermore, SMFs affect the transition metal ions in some enzymes which become paramagnetic, producing an increase in enzyme activity [38].

Previous research has shown that the use of SMF can induce a non-enzymatic defense response [28,39] as carotenoids and in turn, these pigments are known for their industrial applications [39]. The aim of our study was to determine the effect of SMF on the composition of carotenoid pigments in both microalgae. As described by Ambati et al. [28], microalgae of the genus *Scenedesmus* spp. are known to produce β-carotene, lutein, canthaxanthin, astaxanthin, and fucoxanthin. However, the pigment compositions of algae vary from species to species and also according to environmental conditions including different stress conditions such as temperature, salinity, and irradiance which may induce carotenoid production in algal species. Therefore, in our study by U-HPLC analysis, the complete pigment spectrum was determined for both microalgae in the control condition and treatments. Subsequently, using Xcalibur software, a quantitative analysis and identification of the pigments according to the chromatograms was performed. The peaks that showed significant differences between the control and the treatments were detected, and this analysis is shown in Figure 5, only for the pigments that showed differences between the treatments with respect to the control. In *S. obliquus*, the activity in terms of metabolites such as violaxanthin and lutein shows changes below 1%. On the contrary, in *N. gaditana* a significant change was observed in terms of the abundance of violaxanthin, where the increase in production under continuous mode reached 10% and 40% under the pulse exposure mode. This outcome indicates that the production of ROS occurs as a consequence of the oxidative stress in cells caused by the exposure to SMF, coinciding with Guo et al. and Lozano et al. [40,41], where the ROS formation increases the enzymatic activity with respect to the control. In addition, in the future it will be interesting to use pulse SMF exposure as an alternative to increase violaxanthin production as other stress conditions such as temperature, salinity, and irradiance may induce carotenoid production in algal species [42].

## 4. Materials and Methods

### 4.1. Algal Strain and Growth Conditions

The microalgae species, the *S. obliquus* (Fitoplancton Marino S.L., Cadiz, Spain) and the *N. gaditana* CCMP27, were cultured in 1 L volumes. The bold basal medium (BBM) was used for the *S.*
*obliquus* [43,44], which is prepared from fresh water. In the case of *N. gaditana.*, the medium was a modified f/2 [45] prepared from sea water. Both cultures were prepared in glass flasks of 1000 mL with constant air inflow through a tube (see Figure 6). Cultures were incubated for 4 days at 23 ± 1 °C under continuous light using cold-white fluorescent lights (see Figure 7) at 100 μE m^−2^s^−1^ (irradiances were measured with a calibrated quantum sensor and spectroradiometer, LI-250A, LI-COR1800, Li-COR, Biosciences, Lincoln, NE, USA). The initial concentrations of *S. obliquus* and *N. gaditana* in all groups were 1 × 10 ^5^ cells mL^−1^. Every 24 h samples of each culture were collected by centrifugation (10,000× *g*, 10 min at 4 °C) and washed twice with distilled water. Cells were measured at optical density 680 nm (OD_680_) in a Spectroquant Prove 300 spectrophotometer (Merck, Germany). The growth curve for both microalgae were obtained.

### 4.2. Experimental Setup

The setup consists of a flat bottom flask surrounded by an array of neodymium magnets as shown in Figure 8. The neodymium magnets (Nd_2_Fe_14_B of grade N33) are coated with Ni/Cu/Ni and have dimensions of 20 × 20 × 20 mm^3^. The rare earth (RE) content in the RE-Fe alloy of the magnet ranges between 30 and 35 wt%. According to manufacturing specifications, the corresponding remanent flux density $B_{r}$, coercivity values $BH_{c}$ and $jH_{c}$, and maximum stored energy ${\left(BH\right)}_{max}$ are, respectively, 1.164 T, 11.43 kOe and 25.99 kOe, and 33.06 MGOe. The flask contains microalgae media, subject to illumination (see Figure 8). In addition, a thin tube entering from the top was used to introduce air near the bottom. The air inflow results in bubbles which in turn, due to buoyancy, lead to fluid circulation. Regarding the groups, there were 3 flasks for 2 different configurations and 2 exposure modes (continuous and pulse), and additionally, 3 flasks for a control case without SMF exposure (18 flasks for each species). The mode “continuous” means that the biological system was continuously exposed to the SMF during a period of 96 h. The mode “pulse” refers to an intermittent application of the SMF, meaning that the culture was exposed only 1 h per day for 96 h. The latter means that for each microalga, a control condition was considered without exposure to SMF, North-P, North-C, South-P, and South-C in triplicate. The implemented cases under study are enumerated in the following.
Control (C): No magnetic field applied.South-continuous (SC): All south poles are oriented towards the center of the flask and continuous exposure to the magnetic flux density (B).South-pulse (SP): All south poles oriented towards the center of the flask and pulsed exposure to B.North-continuous (NC): All north poles oriented towards the center of the flask and continuous exposure to B.North-pulse (NP): All north poles oriented towards the center of the flask and pulsed exposure to B.


### 4.3. Measurement Procedures

#### 4.3.1. Physicochemical Medium Parameters

Temperature, pH, conductivity (σ), salinity, pressure, and dissolved oxygen were characterized with an HI 98194 multiparameter probe from Hanna Instruments. Samples were taken from the control and treated groups of both media at the exponential growth phase. The measurements were taken every 24 h for a period of 96 h.

#### 4.3.2. Magnetic Flux Density

The magnetic flux density norm was determined for every individual magnet used in all magnet configurations by using an AC/DC magnetic meter composed of a plate-like Hall sensor with a temperature compensation unit. The device is conceived for measurements of homogenous magnetic flux density at the level of the active part (magnet). The field direction of the measurement is uniaxial, sensing the main magnetic component perpendicular to the chip [46], |B|, in gauss (G), and mT units with a resolution of up to 0.1 G in both DC and AC mode and an accuracy of ±5% at 23 ± 5 °C. For good measurements, [47] pointed out that the current density and the magnetic field must be as much orthogonal as possible and that the shape of the active region has to facilitate the measurement of the Hall voltage. Considering measurements of |B| for each configuration, the average values ranged from 4441 to 4547 G with a standard deviation from 0.8% to 3.6% in the worst case. The |B| values at the side of each magnet were used to visualize B in space, which leads to the oxidative stress.

The investigation of the B distribution was performed by means of a mutiphase turbulent bubbly flow coupled with Maxwell equations implemented in COMSOL Multiphysics v5.5. The required inputs such as the remanent magnetic flux density ($B_{r}$) of each magnet was obtained by using the measured magnetic flux density norm |B| at the side of magnets in Equation (1) and solved for $B_{r}$, where L, W, and D are the length, width, and depth of a rectangular magnet [48,49,50]. The computed $\left|B_{r}\right|$ results in values in the range of 1.03 ± 0.02 T. Replacing densities and viscosities [51] at the temperature of the experiment (20 °C) as well as the measured conductivities [30] for both fresh and seawater gives a Reynolds number Re>10,000 [52] and a magnetic Reynolds number $Re_{m}\ll 1$ [53]. Thus, the fluid is turbulent, and B is determined by boundary conditions, not the fluid flow. Thus, the visualization of B can be computed as that field in absence of fluid flow.
(1)$$\left|B\left(d\right)\right|=\frac{B_{r}}{\pi }\left\{tg^{-1}\left(\frac{L\cdot W}{2d\sqrt{4d^{2}+L^{2}+W^{2}}}\right)-tg^{-1}\left(\frac{L\cdot W}{2\left(D+d\right)\sqrt{4{\left(D+d\right)}^{2}+L^{2}+W^{2}}}\right)\right\}$$

#### 4.3.3. Enzymatic Activity

The superoxide dismutase (SOD) and catalase (CAT) total enzymatic activities were measured, since both groups of enzymes are characterized to present paramagnetic elements (Fe, Mn, and Cu) in their catalytic domain, and therefore affected by the magnetic fields in which the algae were exposed during the experiment. The cell break was carried out in a mortar with liquid nitrogen and glass beads. The homogenization was performed with 2 mL of extraction buffer (K_2_HPO_4_-KH_2_PO_4_) KH_2_PO_4_, 50 mm, pH 7.0, containing Triton X-100 (0.01% *v*/*v*) [54]. In the following, the homogenate was centrifuged at 13,000× *g* at 4 °C for 15 min [55]. Proteins were quantified according to the Bradford method [56]. The extract was immediately used to quantify the antioxidant enzymes. The measurement of the SOD activity was measured for both microalgae based on the photochemical reduction of Nitro blue tetrazolium chloride (NBT) described by Donahue et al. [57] and modified by Cartes et al. [55]. For CAT activity, the evaluation was performed through spectrophotometry by the disappearance of hydrogen peroxide according to [58].

#### 4.3.4. Pigments Determination by Ultra High-Performance Liquid Chromatography (UHPLC) Mass-to-Mass

The extracted total pigments were filtered at 0.45 µm and stored at −80 °C until use, according to the protocol described in [59]. The high-performance liquid chromatography (UHPLC) method was used for the quantitative identification of carotenoids as described by Fu et al. [59]. For the separation of carotenoids, a Dionex Ultimate 3000 chromatograph with a diode array detector connected to an Orbitrap Q Exactive Focus (Thermo Scientific, ThermoFisher Scientific, MA, USA) was used. The results were analyzed using Thermo Xcalibur Sequence Setup software (Thermo Scientific, ThermoFisher Scientific, Waltham, MA, USA).

#### 4.3.5. Statistical Analysis

The data of the different treatments were analyzed with univariate statistical models using analysis of variances (ANOVA) with a confidence of 95%. These analyses were performed with the Minitab 17 software.

## 5. Conclusions

In this study we evaluate the response of two different microalgae to the presence of static magnetic fields (SMF) exposed in continuous and pulsed form. The SMFs used have north and south configurations. For both microalgae, *S. obliquus* and *N. gaditana*, a clear negative effect on growth was observed in the presence of the north configuration, both in pulse and continuous exposure, after 48 h of culture. Other effects were evidenced in the alteration of antioxidant enzyme activity in SOD and CAT enzymes. In *S. obliquus*, an over-regulation of SOD in the presence of SMF in continuous exposure and a down-regulation in pulse exposure to SMF were observed for both the north and south configurations. In contrast, for CAT only an increase in enzyme activity is observed only in continuous exposure to SMF. In the case of *N. gaditana*, an over-regulation of SOD and CAT enzyme activity is observed upon continuous and pulse exposure to SMF in both north and south configurations. The latter coincides with a significant increase in antioxidant pigments in *N. gaditana*, especially violaxanthin, which is part of the non-enzymatic defense system, but may have an interesting use as a bioactive molecule, which in future studies could be scaled up and optimized using SMF.

## Figures and Tables

**Figure 1 marinedrugs-19-00527-f001:**
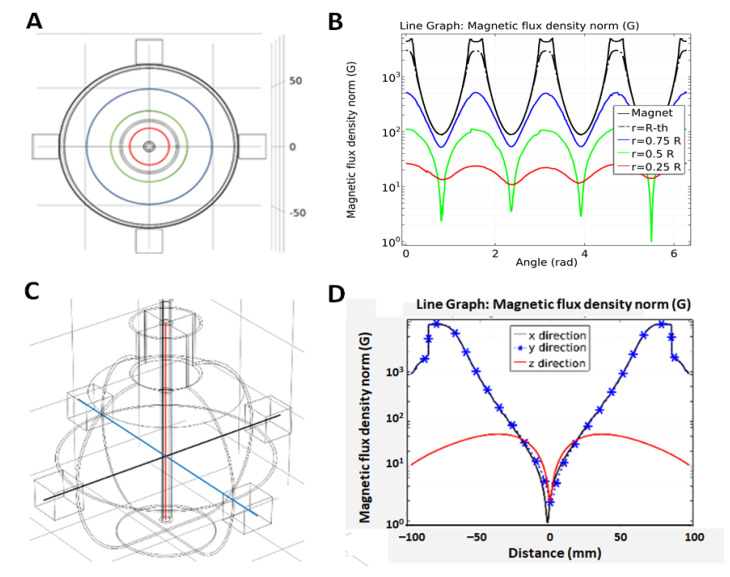
(**A**) Top view indicating along the curve where the magnetic flux density norm |B| is evaluated. (**B**) Plot of |B| where all north poles are oriented towards the center of the flask, evaluated along curves shown in (**A**). (**C**) Lateral view to specify B field along x, y, and z directions. (**D**) Plot of |B|, evaluated along curves shown in (**C**).

**Figure 2 marinedrugs-19-00527-f002:**
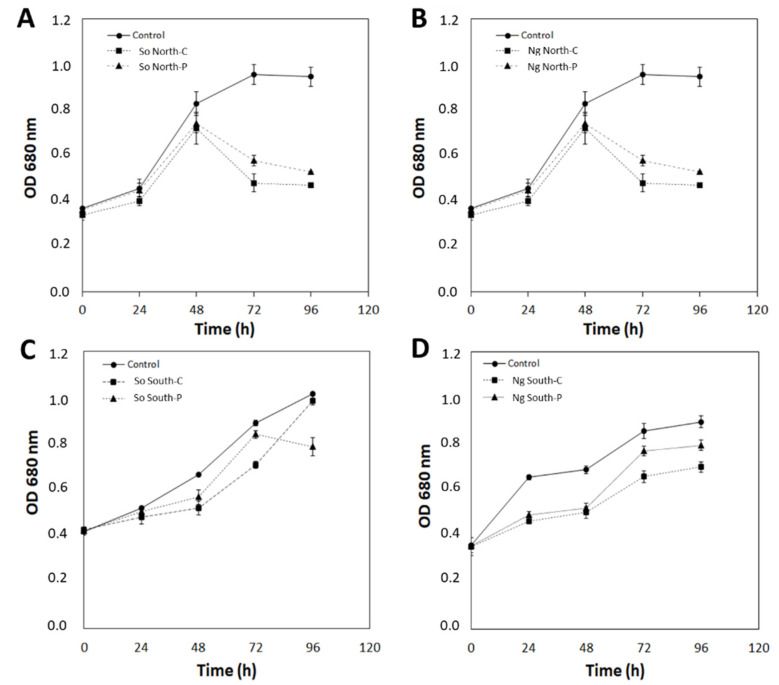
Growth kinetics of *S. obliquus* and *N. gaditana* in response to pulsed and continuous SMF exposure. North arrangement for *S. obliquus* (**A**) and for *N. gaditana* (**B**). South configuration for *S. obliquus* (**C**) and for *N. gaditana* (**D**). Data are presented as mean ± standard deviation analyzed from three replicates. Square corresponds to the control without exposure to SMF. So: *S. obliquus*; Ng: *N. gaditana*; North-C: north-continuous exposure; North-P: north-pulse-type exposure; South-C: south-continuous exposure; South-P: south-pulse-type exposure.

**Figure 3 marinedrugs-19-00527-f003:**
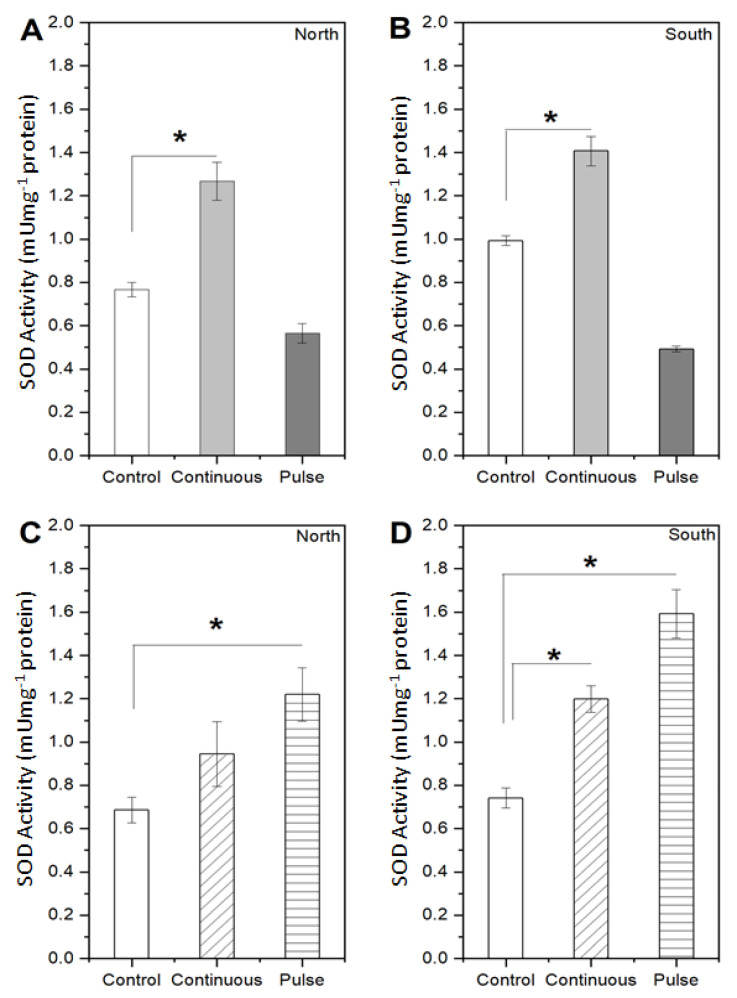
Measurements of the enzymatic activity in terms of SOD for the two microalgae unexposed or control and exposed to SMF with north and south configurations under continuous and pulse exposure modes. (**A**) North and (**B**) south magnet configuration for the *S. obliquus;* (**C**) north and (**D**) south magnet configuration for the *N. gaditana*. Data represent the average of three replicates and the standard deviation (±). Significant differences between the control and experimental treatments with univariate statistical models using ANOVA analysis with a confidence of 95% (*p* values < 0.05 are indicated by *).

**Figure 4 marinedrugs-19-00527-f004:**
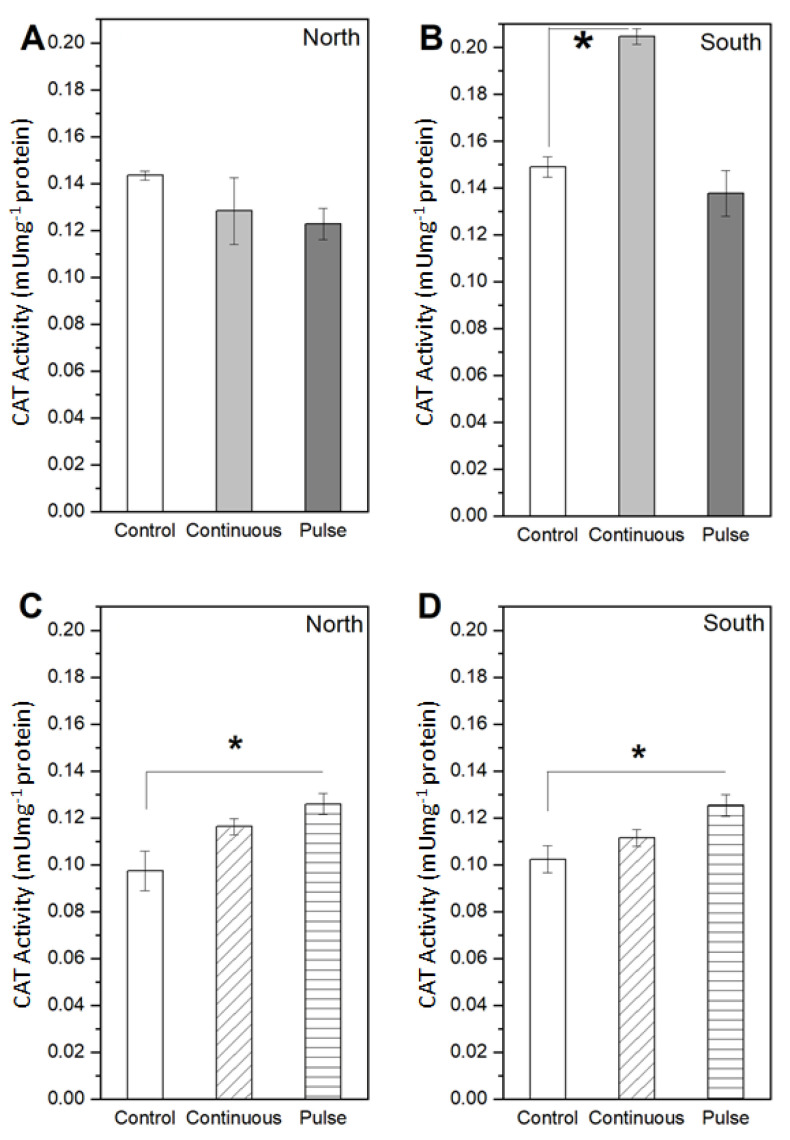
Measurements of the enzymatic activity in terms of CAT for the two microalgae unexposed or control and exposed to SMF with north and south configurations under continuous and pulse exposure modes. (**A**) North and (**B**) south magnet configuration for the *S. obliquus*. (**C**) North and (**D**) south magnet configuration for the *N. gaditana*. Data represent the average of three replicates and the standard deviation (±). Significant differences between the control and experimental treatments with univariate statistical models using ANOVA analysis with a confidence of 95% (*p* values < 0.05 are indicated by *).

**Figure 5 marinedrugs-19-00527-f005:**
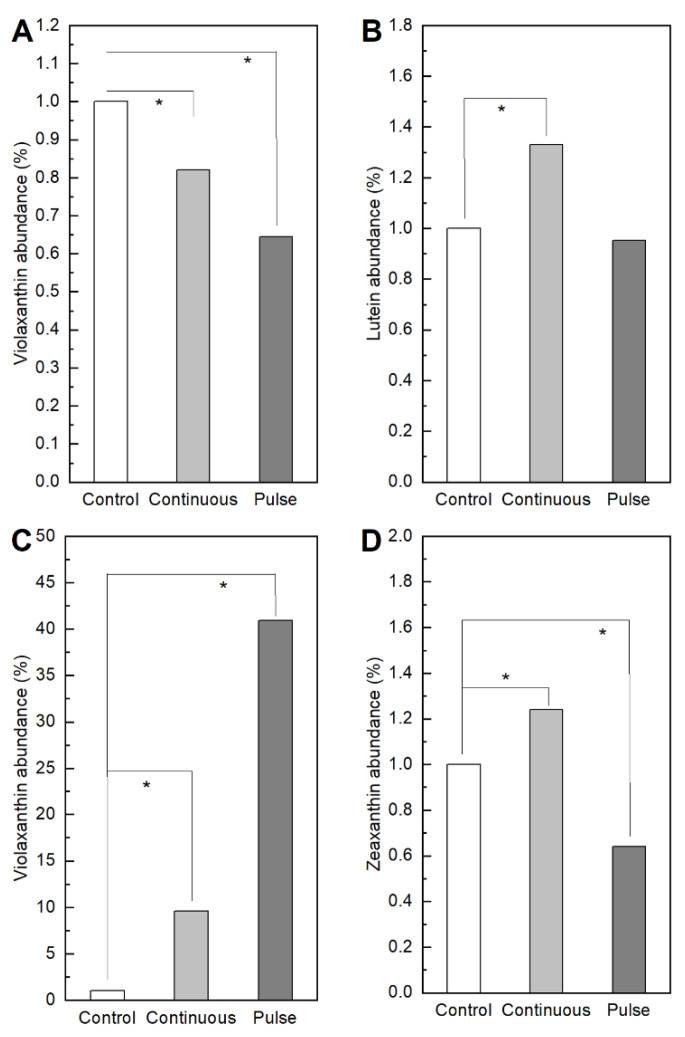
Quantification of metabolites of interest. (**A**,**B**) show the abundance of violaxanthin and lutein for *S. obliquus*; while (**C**,**D**) correspond to the abundance of violaxanthin and zeaxanthin for *N. gaditana*. Data represent the average of three replicates and the standard deviation (±). Significant differences between the control and experimental treatments with univariate statistical models using ANOVA analysis with a confidence of 95% (*p* values < 0.05 are indicated by *).

**Figure 6 marinedrugs-19-00527-f006:**
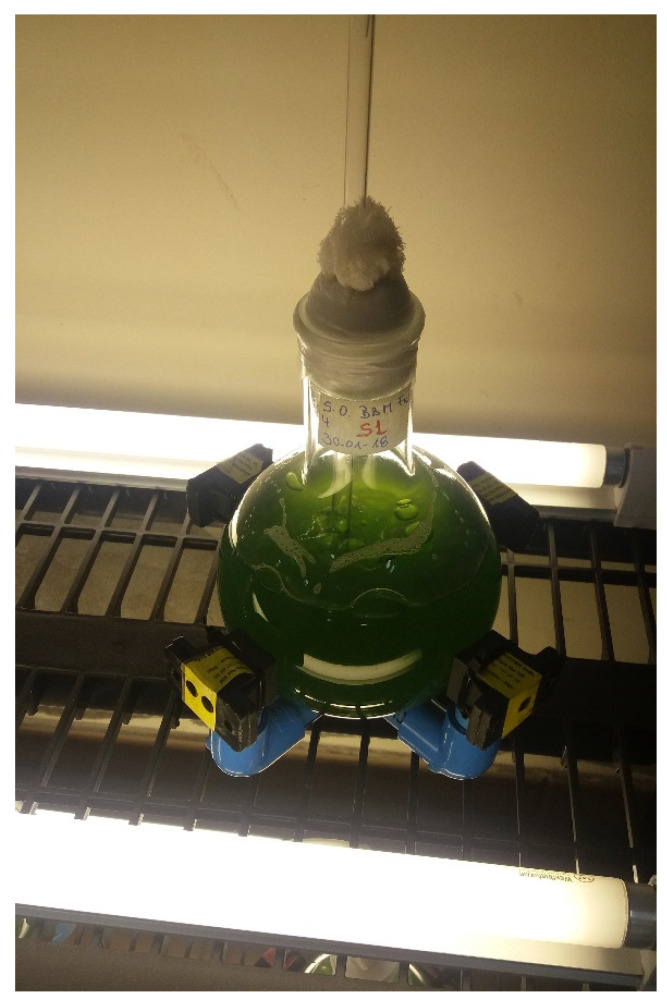
Photograph of the experimental setup for a single flask having the culture inside, surrounded by the permanent magnets, and under illumination.

**Figure 7 marinedrugs-19-00527-f007:**
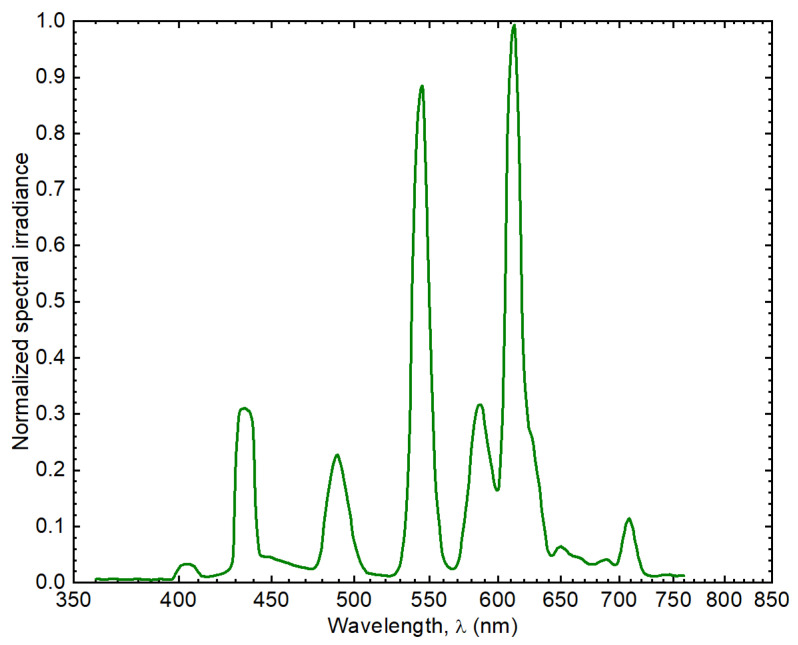
Spectrum of the light source. This spectrum is characterized by sharp peaks of intensities near the maximum magnitude at 545 nm and 613 nm, and several ones of intensities below 30% of the maximum at 435 nm, 489 nm, 587 nm, and 709 nm.

**Figure 8 marinedrugs-19-00527-f008:**
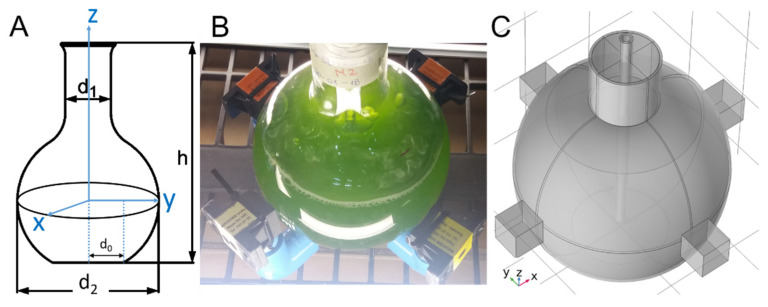
(**A**) Geometrical measures of the flask: $d_{0}=30\;\mathrm{mm}$, $d_{1}=42\;\mathrm{mm}$, $d_{2}=131\;\mathrm{mm}$, and h=190 mm. (**B**) Glass flask surrounded by an array of permanent magnets at z=0 and a thin tube to introduce air. (**C**) Geometry model in 3D. The neodymium magnets (Nd_2_Fe_14_B of grade N33) are coated with Ni/Cu/Ni and have dimensions of 20 × 20 × 20 mm^3^. The rare earth (RE) content in the RE-Fe alloy of the magnet ranges between 30 and 35 wt%. According to manufacturing specifications, the corresponding remanent flux density $B_{r}$, coercivity values $BH_{c}$ and $jH_{c}$, and maximum stored energy ${\left(BH\right)}_{max}$ are, respectively, 1.164 T, 11.43 kOe and 25.99 kOe, and 33.06 MGOe.

**Table 1 marinedrugs-19-00527-t001:** Summary of the physicochemical medium parameters for *S. obliquus* and *N. gaditana.* Averages and standard deviations were computed as each magnet configuration and control were implemented in triplicate. NC: north-continuous; NP: north-pulse; SC: south-continuous; SP: south-pulse.

Species	Treatment	T (°C)	pH	σ (S/m)	S (PSU)	DO (%)
*S. obliquus*	Control for North	21.1 ± 2.3	8.7 ± 0.8	0.09 ± 0.01	0.47 ± 0.07	6.5 ± 0.9
	NC	21.0 ± 2.4	8.5 ± 0.8	0.1 ± 0.01	0.49 ± 0.05	6.8 ± 0.9
	NP	21.0 ± 2.5	9.0 ± 1.1	0.09 ± 0.01	0.46 ± 0.08	6.6 ± 1.1
	Control for South	22.8 ± 0.3	8.9 ± 0.5	0.10 ± 0.01	0.48 ± 0.05	5.7 ± 0.3
	SC	23.0 ± 0.5	8.6 ± 0.7	0.09 ± 0.03	0.42 ± 0.14	5.9 ± 0.6
	SP	23.0 ± 0.8	8.8 ± 0.8	0.09 ± 0.02	0.47 ± 0.11	6.0 ± 0.6
*N. gaditana*	Control for North	21.2 ± 1.4	8.4 ± 0.4	3.7 ± 0.4	23.7 ± 2.5	5.2 ± 0.3
	NC	21.3 ± 1.8	8.6 ± 0.2	4.4 ± 0.6	28.5 ± 4.6	5.4 ± 0.3
	NP	21.3 ± 1.5	8.6 ± 0.2	4.4 ± 0.6	28.9 ± 4.5	5.2 ± 0.4
	Control for South	21.1 ± 0.2	8.6 ± 0.2	4.2 ± 0.9	27.5 ± 6.3	5.0 ± 0.3
	SC	21.3 ± 0.1	8.6 ± 0.2	4.3 ± 0.6	28.2 ± 4.1	5.1 ± 0.2
	SP	21.2 ± 0.2	8.6 ± 0.2	5.0 ± 0.6	33.0 ± 4.4	4.9 ± 0.3

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
