# Peer review of "Response to Static Magnetic Field-Induced Stress in *Scenedesmus obliquus* and *Nannochloropsis gaditana"

_marinedrugs, 2021, doi:10.3390/md19090527_

Round 1

Reviewer 1 Report

The manuscript “Response to static magnetic field-induced stress in Scenedesmus obliquus and Nannochloropsis gaditana” describes effect of the magnetic field on two un-related algal species. The findings in the MS might provide further information about effect of the magnetic field on cells in general and algae specifically. There is also a potential for inducing production of carotenoids such as viaolaxanthin, which would make the findings interesting for algal biotechnology. Yet, the manuscript suffers several major flaws as well as some minor issues.

Major issues:

  • The algae were grown in sub-optimal conditions since according to the OD750 data the cells of obliquus only doubled within 48 hours. In optimal conditions, the species is able to increase the cell size and cell numbers 8-fold in 24 hours. It is not possible to draw any significant conclusions from such sub-optimal conditions. Although it is possible that the effect of magnetic field is only visible on already stressed cells in such sub-optimal conditions, this should be made clear by comparing the effect to optimal conditions. On the same point, it is also required to provide more details about the cultivation, such as: How was the optical density at 750 nm measured? What was the distance of the cuvette? What were the cell numbers during the cultivation? For more points see also below in the Minor points.
  • The manuscript presents no proof that ROS production is being induced by magnetic field although it is assumed throughout the MS and specifically claimed in Discussion (l. 311-314). To show the production of ROS in the cultures optimally a combination of different methods should be used. The standardly used methods include measuring lipid peroxidation, establishing survival rates, and direct fluorescent labeling of ROS within the cells.
  • What was the rationale for quantification of violaxanthin, zeaxanthin and lutein? In obliquus both violaxanthin and lutein represent minor pigments, which was also proved by the authors. So it is to be expected that the differences will be low, below 1 % as stated. Were some other carotenoids quantified? Was there some difference?

Minor points:

  • The Introduction is too long and too detailed in places.
  • l. 107 Please put reference Small et al. 2012 into correct format
  • l. 109-110 The sentence „Moreover it has been…“ is unclear, please re-phrase it.
  • What is the rationale of studying ROS production in the presence of magnetic field? Please describe it in the introduction.
  • Figure 1 was inserted twice.
  • Figure 2 - add numbers to Y axis in B to make the figure easier to read.
  • Please identify the statistical test used within the Figure legends.
  • Figure 3 Unify the font size of Y axis labels in A with the rest of the figure.
  • Figure 5 Unify the presentation of the statistical significance with figures 3 and 4.
  • l. 259-263 I would suggest to move this rationale to introduction.
  • l. 324 What was the volume of the culture used?
  • l. 326 Where was the light intensity measured? Is it incident light intensity? At which part of the vessel? State the shape of the vessel.
  • l. 410 How was the method modified? Please describe the details.

Reviewer 2 Report

Dear authors.

Algae and microalgae attract the attention of researchers. The topic touched upon in the article is relevant. The scientific content of the manuscript justifies its publication, but some additions and a significant revision of the material are required.

Major comments:

1) Introduction should be reduced and rewritten.

2) The choice of algae S. obliquus and N. gaditana must be justified

3) L. 106-107, the link format should be changed.

4) L. 243-244 should be rewritten.

5) «The * symbol stands for statistical significance p<0.005 with 95% confidence». The sentence is not clear, you need to rewrite it

6) Statistical analysis is not described enough.

7) In Сonclusion you need to add practical recommendations for using the obtained regularities, rather than listing the results

8) In the References, 14% of publications refer to 2017-2021 (the last 5 years); the remaining 86% of used sources are older than 5 years. It is recommended to increase the share of references to sources published over the last 5 years when analyzing the current state of research in the area under consideration, since this area of knowledge is rapidly developing.

Round 2

Reviewer 1 Report

The manuscript has improved thanks to incorporation of the reviewer´s comment. Yet, some of the comments, suggestions and concerns raised by this reviewer have only been reacted upon in the response to reviewer file but are not reflected within the manuscript itself. Their point was to clarify the manuscript so the responses should appear there.

This applies to one of the major points, the justification of the quantification of the carotenoids. The reader should know why they were quantified and why were the ones presented chosen. Simply the text from the response to reviewer could be used.

Also, there is still not enough information on the illumination of the vessels used for the cultivation. The shape of the vessel and the place where the light intensity was measured is important for understanding the growth of the algae and it should be specified. Also, it is not clear what was the light source and consequently the light distribution? All the details of the cultivation set up are required for the reader to be able to evaluate the significance of the data.

In reaction to response to the comment 1.

  • Period of 4 days is not so “short” period of time as it, in more favorable conditions, corresponds to four generations of algae. So in fact the authors are assessing not an immediate response but acclimation to the presence of magnetic field. This is perfectly fine; it just should be kept in mind for the interpretation.
  • Photosynthetic efficiency serves to detect stress. Yet, there is wide range of growth conditions, which are not stressful to the cells but they will significantly affect the growth and behavior of the algae. This makes the detailed description of the growth conditions a crucial part of the MS.

Reviewer 2 Report

Dear Autors. My comments are taken into account.

Round 3

Reviewer 1 Report

The comments were incorporated into the MS. I only have one further comment. I would introduce the photograph of the cultivation vessel and the graph of the light source which was provided in response to reviewers into the MS as a new Figure 6 and make the current Figure 6 into Figure 7.
